# Klp10A, a stem cell centrosome-enriched kinesin, balances asymmetries in *Drosophila* male germline stem cell division

Cuie Chen†, Mayu Inaba†, Zsolt G Venkei†, Yukiko M Yamashita*

Department of Cell and Developmental Biology, Life Sciences Institute, Howard Hughes Medical Institute, University of Michigan, Ann Arbor, United States

**Abstract** Asymmetric stem cell division is often accompanied by stereotypical inheritance of the mother and daughter centrosomes. However, it remains unknown whether and how stem cell centrosomes are uniquely regulated and how this regulation may contribute to stem cell fate. Here we identify Klp10A, a microtubule-depolymerizing kinesin of the kinesin-13 family, as the first protein enriched in the stem cell centrosome in *Drosophila* male germline stem cells (GSCs). Depletion of *klp10A* results in abnormal elongation of the mother centrosomes in GSCs, suggesting the existence of a stem cell-specific centrosome regulation program. Concomitant with mother centrosome elongation, GSCs form asymmetric spindle, wherein the elongated mother centrosome organizes considerably larger half spindle than the other. This leads to asymmetric cell size, yielding a smaller differentiating daughter cell. We propose that *klp10A* functions to counteract undesirable asymmetries that may result as a by-product of achieving asymmetries essential for successful stem cell divisions.

*For correspondence: yukikomy@umich.edu

†These authors contributed equally to this work

## Introduction

Asymmetric cell division (ACD) is a key process that balances stem cell self-renewal and differentiation by producing one stem cell and one differentiating cell (*Inaba and Yamashita, 2012*; *Morrison and Kimble, 2006*). ACD is also critical for generating diverse cell types during embryonic development. During ACD, many cellular components have been reported to segregate asymmetrically, including fate determinants, certain organelles, sister chromatids/histones and damaged proteins (*Chen et al., 2016*; *Tajbakhsh and Gonzalez, 2009*). Although cellular asymmetries are critical aspects of ACD, the essence of successful cell division is the precise replication and segregation of cellular contents, such as chromosomes and essential organelles. It has been underexplored how cells may achieve productive ACD without interfering with the basic requirement of cell divisions.

The stereotypical inheritance of mother and daughter centrosomes during ACD has been observed in several stem cell systems (*Conduit and Raff, 2010*; *Habib et al., 2013*; *Januschke et al., 2011*, *2013*; *Rebollo et al., 2007*; *Rusan and Peifer, 2007*; *Wang et al., 2009*). As the major microtubule-organizing centers in the cell, the centrosomes have the ability to influence the segregation of many cellular components during cell division: most critically, the centrosomes play a fundamental role as spindle poles during mitosis to achieve faithful, equal segregation of sister chromatids (*Meraldi, 2016*). On the other hand, it is reported that aggresomes (damaged protein aggregates) are segregated asymmetrically to only one cells during cell division, by being associated with a centrosome (*Rujano et al., 2006*). It is also reported that proteins destined for degradation are targeted to the centrosomes and segregated asymmetrically during human

embryonic stem cell divisions (*Fuentealba et al., 2008*). We have shown that biased segregation of sister chromatids (*Yadlapalli and Yamashita, 2013*) and asymmetric inheritance of the midbody ring (*Salzmann et al., 2014*) depend on the centrosomes in *Drosophila* male germline stem cells (GSCs). In these examples, it is plausible that a slight difference in MTOC activities between the mother and the daughter centrosomes is utilized to achieve asymmetric segregation of cellular components (*Tajbakhsh and Gonzalez, 2009*; *Yamashita et al., 2007*). This raises two critical questions: (1) how might the mother and daughter centrosomes be distinctly regulated within the stem cells to achieve asymmetric segregation of certain cellular components? And (2) how do the mother and daughter centrosomes achieve the critical balance of segregating certain cellular components asymmetrically, while achieving equal segregation of other components such as sister chromatids?

Asymmetric centrosome inheritance was first described in *Drosophila* male GSCs, where the mother centrosome is consistently inherited by the stem cell while the daughter centrosome is inherited by the differentiating cell (*Yamashita et al., 2007*). ACD of GSCs is influenced by their stem cell niche, which secretes self-renewal ligands and specifies GSC identity. GSCs attach to the hub cells, which function as a major niche component (*Figure 1A*). The mother centrosomes in the GSCs are positioned near the hub cells throughout the cell cycle, whereas the daughter centrosomes migrate toward the distal side of the GSCs, and are inherited by the gonialblasts (GBs), the differentiating daughters of GSCs (*Yamashita et al., 2003*, *2007*)(*Figure 1A*). Such stereotypical positioning of centrosomes helps orient the mitotic spindle perpendicular to the hub cells, leading to retention of GSCs within the niche and displacement of GBs away from the niche and ensuring the asymmetric outcome of the stem cell divisions. Although it is clear that this stereotypical centrosome positioning would help orient the mitotic spindle to achieve ACD, many intriguing questions surrounding the phenomenon of asymmetric centrosome inheritance remain to be answered: Do the mother and/or

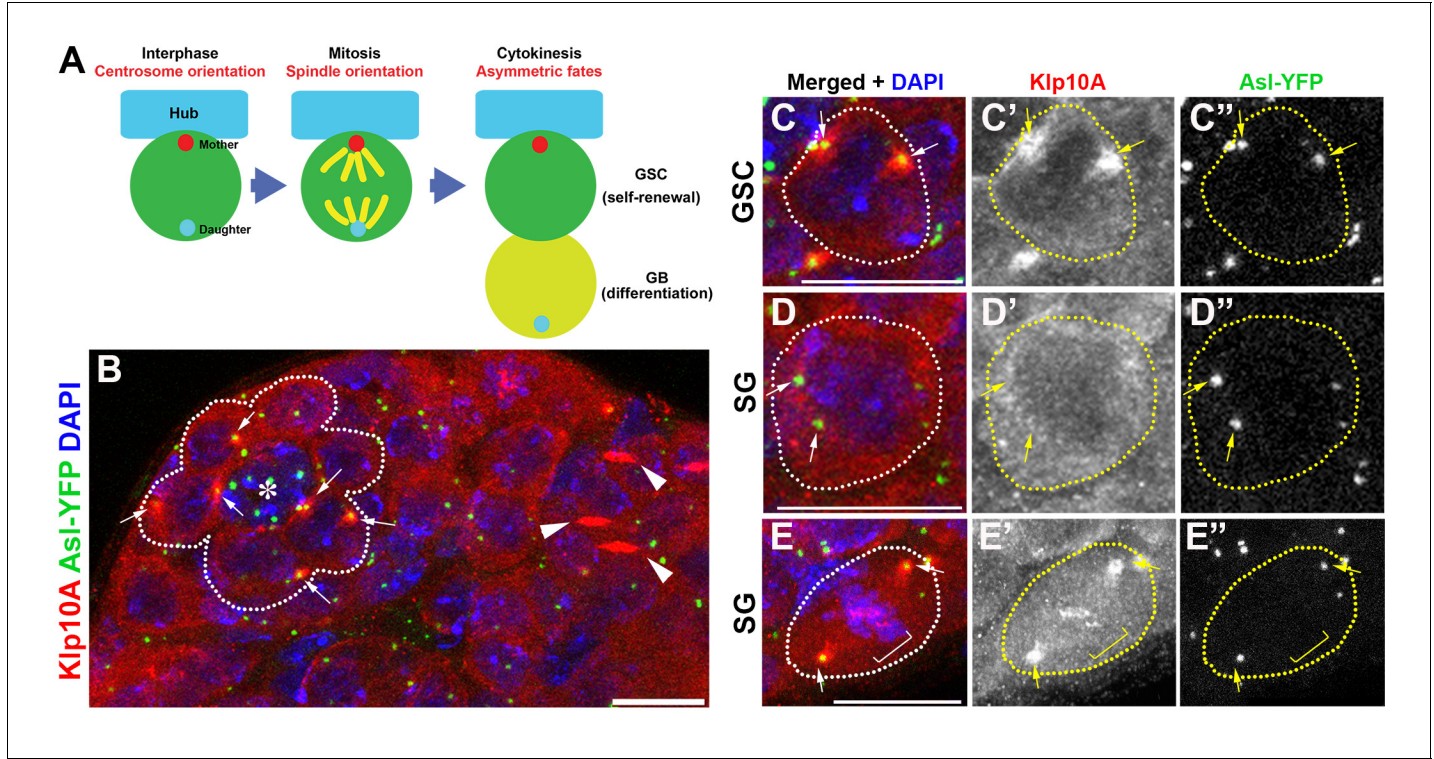

**Figure 1.** Klp10A localizes to the GSC centrosomes. (**A**) Centrosome behavior in male GSCs. (**B**) An apical tip of the testis stained for Klp10A (red), Asl-YFP (green, centrosome) and DAPI (blue). Arrows indicate GSC centrosomes. GSCs are indicated by broken lines. Arrowheads indicate Klp10A localization to the central spindle microtubule bundle. Hub (*). Bar: 10 μm. (**C**, **D**) magnified images of an interphase GSC (**C**) and an interphase spermatogonium (**D**) chosen from panel B, demonstrating that Klp10A localization to the interphase centrosome is specific to GSCs. **E**) Mitotic spermatogonium showing Klp10A localization to the spindle poles (arrows) and kinetochores (bracket). Bar: 5 μm. It should be noted that MT-nanotubes are sensitive to regular fixation conditions, and its localization on MT-nanotubes is not obvious in the images shown in panel B.

daughter centrosomes carry fate determinants or any characteristics that contribute to fate determination? Are stem cell centrosomes regulated differently than the centrosomes in non-stem cells?

In this study, we identify Klp10A, a member of the kinesin-13 family of microtubule-depolymerizing motors, as the first protein described to date that is enriched on centrosomes specifically in the GSCs. We show that depletion of *klp10A* results in elongation of the mother centrosomes in stem cells, but not the daughter centrosomes in the stem cells or any centrosomes in SGs, differentiating progeny of stem cells. Elongated mother centrosomes in GSCs result in multiple undesirable asymmetries, such as sister chromatid missegregation of small fourth chromosomes and asymmetric daughter cell size, suggesting that *klp10A* is required to balance asymmetries during ACD. Based on our findings, we propose that the mechanisms of ACD have the potential to generate undesirable asymmetries, which must be counterbalanced to achieve functional/productive ACD.

## Results

### Klp10A is enriched at the GSC centrosomes

In the course of our previous study characterizing microtubule-based nanotubes (MT-nanotubes) (*Inaba et al., 2015*), we found that Klp10A localizes to the interphase centrosomes specifically in GSCs (*Figure 1B*, arrows). Klp10A is a member of the kinesin-13 family of microtubule-depolymerizing motors, playing roles in regulation of primary cilia (*Kobayashi et al., 2011*) and MT-nanotubes (*Inaba et al., 2015*) and in bipolar spindle formation (*Rogers et al., 2004*). Whereas Klp10A localizes to the GSC centrosomes throughout the cell cycle (*Figure 1C*, arrows), its localization to the centrosomes of spermatogonia (SGs), the differentiating progeny of GSCs, is limited to mitosis (*Figure 1D, E*). In both GSCs and SGs, Klp10A was observed on mitotic spindle poles, kinetochores (*Figure 1E* arrows and brackets) and central spindle (*Figure 1B*, arrows), a localization previously shown in cultured cells (*Rogers et al., 2004*). To our knowledge, this is the first protein reported to date to be enriched on the stem cell centrosomes compared to the centrosomes of non-stem cells. These data indicate that Klp10A might play a role in the stem cell-specific function of centrosomes, prompting us to further study the role of Klp10A in GSC divisions.

### Klp10A regulates GSC mother centrosome length

To investigate the role of Klp10A, we used previously validated RNAi-mediated knockdown of *klp10A* in germ cells (*nos-gal4>UAS-klp10A$^{TRiP.HMS00920}$*, referred to as *klp10A$^{RNAi}$* hereafter, *Figure 2—figure supplement 1A*)(*Inaba et al., 2015*) as well as the *klp10A$^{ThbA}$* hypomorphic mutant (*Delgehyr et al., 2012*). In wild-type GSCs, the mother and daughter centrosomes are equal in size with the mother centrosome located near the hub throughout the cell cycle (*Figure 2A*) (*Yamashita et al., 2003*, *2007*). However, in *klp10A$^{RNAi}$* or *klp10A$^{ThbA}$* testes, the centrosomes proximal to the hub cells (presumably the mother centrosomes) in the GSCs elongated dramatically, reaching up to ~7 μm (average of 1.7 ± 1.1 μm, compared to an average of 0.56 ± 0.2 μm in control GSCs, N = 50 GSCs for each genotype), whereas the distal centrosomes (presumably the daughter centrosomes) in the GSCs were slightly shorter than the control centrosomes (*Figure 2B,C*). 46.6% of GSCs examined contained a visibly elongated centrosome (n = 277, *Figure 2B,C*, and *Figure 2—figure supplement 1B*). Transmission electron microscopy confirmed that it was indeed an elongated centrosome, not a primary cilium, because it lacked the surrounding membrane found on primary cilia (*Figure 2—figure supplement 1C,C'*). Elongated centrosomes were positive for three centrosomal markers examined along their entire length (*Figure 2—figure supplement 1D,E*), further confirming their identity as centrosomes. Centrosomes in SGs were not elongated (*Figure 2C*), suggesting that Klp10A's role in preventing centrosome elongation is unique to GSCs. The centrosomes in spermatocytes in *klp10A$^{ThbA}$* hypomorphic mutant were slightly elongated, as reported previously (*Delgehyr et al., 2012*) (*Figure 2C*). However, in spermatocytes, both mother and daughter centrosomes showed the same extent of elongation, in stark contrast to GSCs where only one centrosome is elongated. The centrosome elongation in *klp10A$^{RNAi}$* was rescued by the introduction of an RNAi-insensitive *UAS-GFP-klp10A* transgene (*nos-gal4>UAS-GFP-klp10A*), excluding the possibility of an off-target effect (not shown).

We assumed that the elongated centrosome is the mother, based on its location near the hub, where the mother centrosome is located (*Yamashita et al., 2007*) (*Figure 2B*; In 78% of cases, the

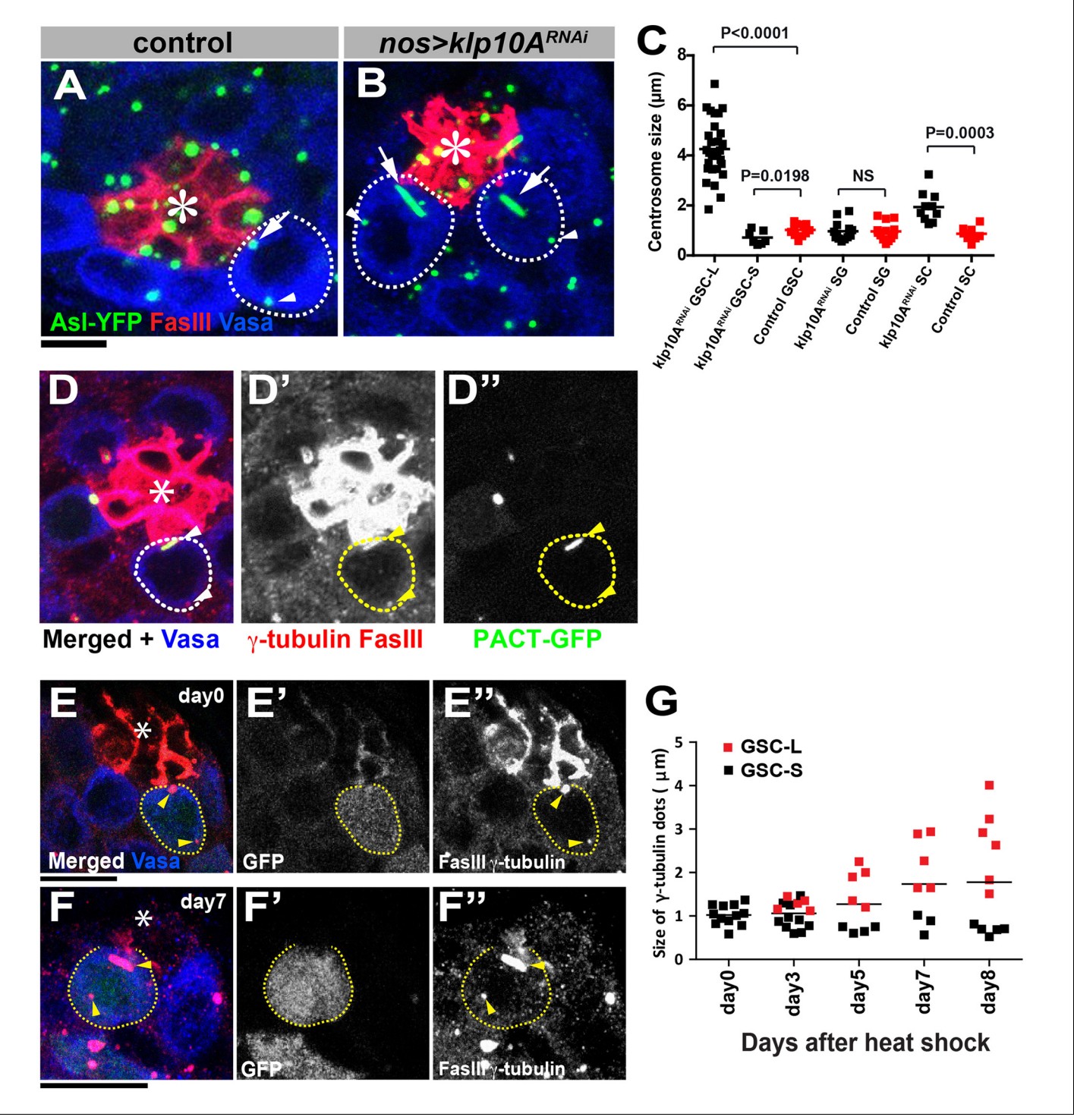

**Figure 2.** *klp10A* is required for maintenance of mother centrosome size in GSCs. Wild-type (**A**) and *klp10A^{RNAi}* (**B**) GSCs stained for Asl-YFP (green, centrosome), Fas III (red, hub), Vasa (blue). GSCs are indicated by dotted circles. Arrows indicate proximal (mother) centrosomes; arrowheads indicate distal (daughter) centrosomes. Hub (\*). Bar: 5 µm. (**C**) Quantification of centrosome size in control (N = 13) and *klp10A^{RNAi}* (N = 30 GSC-L, N = 15 GSC-S) GSCs, control (N = 14) and *klp10A^{RNAi}* (N = 15) SGs, and control (N = 9) and *klp10A^{RNAi}* (N = 11) spermatocytes (SC). (**D**) PACT-GFP (*NGT40*>PACT-GFP) labels the elongated, proximal centrosome. γ-tubulin and Fas III (red), Vasa (blue), GFP-PACT (green). Hub (\*). GSC is indicated by broken line, and the centrosomes are indicated by arrowheads. (**E**, **F**) Examples of *klp10A^{RNAi}* GSC clones at 0 days (**E**) and seven days (**F**) after clone induction (hs-FLP, *nos*>stop>gal4, UAS-GFP, UAS-*klp10A^{RNAi}*). GFP (green, *klp10A^{RNAi}* GSC clones), Fas III, γ-tubulin (red). Bar: 10 µm. (**G**) Quantification of centrosome length in *klp10A^{RNAi}* GSC clones on indicated days after clone induction. Day0 (N = 13), Day3 (N = 16), Day5 (N = 10), Day7 (N = 9) and

*Figure 2 continued on next page*

*Figure 2 continued*

Day8 (N = 12) were scored. Statistical analysis and graphing were performed using GraphPad prism six software. The p value (two-tailed Student's t-test) is provided for comparison between indicated columns in C. Bars in G indicate means. Note that GSC-L and –S indicate longer and shorter centrosomes, respectively, from each GSC in C and G. Scoring of GSC centrosome size in these panels come only from GSCs whose centrosomes shows clear size difference (overall average size including GSCs without visibly elongated centrosomes is given in the text).

The following figure supplement is available for figure 2:

**Figure supplement 1.** Loss of Klp10A leads to mother centrosome elongation.

elongated centrosomes were near the hub, n = 222 GSCs with elongated centrosomes). This notion was further supported by using a mother-centrosome–labeling method reported previously (*Yamashita et al., 2007*). In this method, flies express PACT-GFP under control of the *NGT40-gal4* driver, which preferentially labels the mother centrosome. We found that the elongated centrosomes were preferentially labeled by PACT-GFP (*Figure 2D*): 73% of elongated centrosomes in *klp10A^{RNAi}* GSCs were labeled by PACT-GFP (n = 40/55), whereas the shorter centrosome was labeled by PACT-GFP in 0% of GSCs. In the remaining 27% of cases (n = 15/55), both centrosomes were similar in size and the correlation between the centrosome age (mother/daughter) and length could not be determined. Although we cannot exclude the possibility that PACT is preferentially incorporated into the larger centrosome (irrespective of age) in *klp10A^{RNAi}* GSCs, the above data indicate that it is likely the mother centrosome that elongates upon loss of *klp10A* function.

Interestingly, after the induction of *klp10A^{RNAi}* GSC clones (*hs-FLP; nos>stop>gal4; UAS-GFP, UAS-klp10A^{RNAi}*), the centrosomes elongated gradually, reaching ~3–4 µm in length after eight days from the normal size of 0.56 ± 0.2 µm, *Figure 2E–G*). This result suggests that the centrosome that stays in the GSC (i.e. the mother centrosomes) continuously and gradually elongates. In contrast, the GSC daughter centrosomes do not elongate even after they mature to become the mother centrosome once in differentiating germ cells (GBs/SGs). These results reveal that the GSC mother centrosomes have an inherent tendency to elongate, which is counteracted by the function of Klp10A. This might be because of the need of the GSC mother centrosomes to be constantly anchored to the hub-GSC junction (*Yamashita et al., 2007*), necessitating stronger microtubule nucleating activity throughout the cell cycle. Despite the striking effect of *klp10A* loss of function on centrosome length, overexpression of Klp10A did not result in any effect on the centrosome length (0.69 ± 0.26 µm in control GSCs, 0.71 ± 0.22 µm in *klp10A^{RNAi}* GSCs (N = 67 GSCs for each genotype, p=0.70)).

## Klp10A depletion leads to asymmetric microtubule organization in GSCs

Concomitant with mother centrosome elongation, the two centrosomes in *klp10A^{RNAi}* GSCs showed considerable asymmetry in their microtubule-organizing center (MTOC) activity. In control testes, the mother centrosome organizes slightly more MTs than the daughter in early stages of the cell cycle (*Yamashita et al., 2007*), but the difference diminishes as GSCs reached late G2 phase (*Figure 3A*). Such symmetric MTOCs led to symmetric morphology of the spindle formed during mitosis (*Figure 3B*, 100%, n = 20). In contrast, in *klp10A^{RNAi}* GSCs, the two centrosomes showed dramatic asymmetry in MTOC activity. In interphase, cytoplasmic microtubules emanating from the elongated centrosome were considerably more robust than those emanating from the shorter centrosome within the same cell (*Figure 3C* 64%, n = 28 G2 GSCs). In mitosis, asymmetric MTOC activities between the long and short centrosomes resulted in asymmetric spindle morphology, with the proximal spindle pole organizing considerably more spindle microtubules (*Figure 3D*, 54%, n = 20 mitotic GSCs).

Asymmetric spindles appear to generate asymmetric forces, because we often observed that both sister chromatids of the fourth chromosomes, identified by its small size, segregated toward the larger centrosome. It has been documented that the sister chromatids of chromosome 4, easily identified by its small size, segregate to the spindle poles before the onset of anaphase and are visualized as phosphor-histone H3 (PH3)-positive dots near the spindle poles during metaphase (*Dej et al., 2004*; *Orr-Weaver, 1995*). In control metaphase cells, PH3-positive chromosome 4 s were always observed at both spindle poles, and missegregation was never observed (0%, n = 14

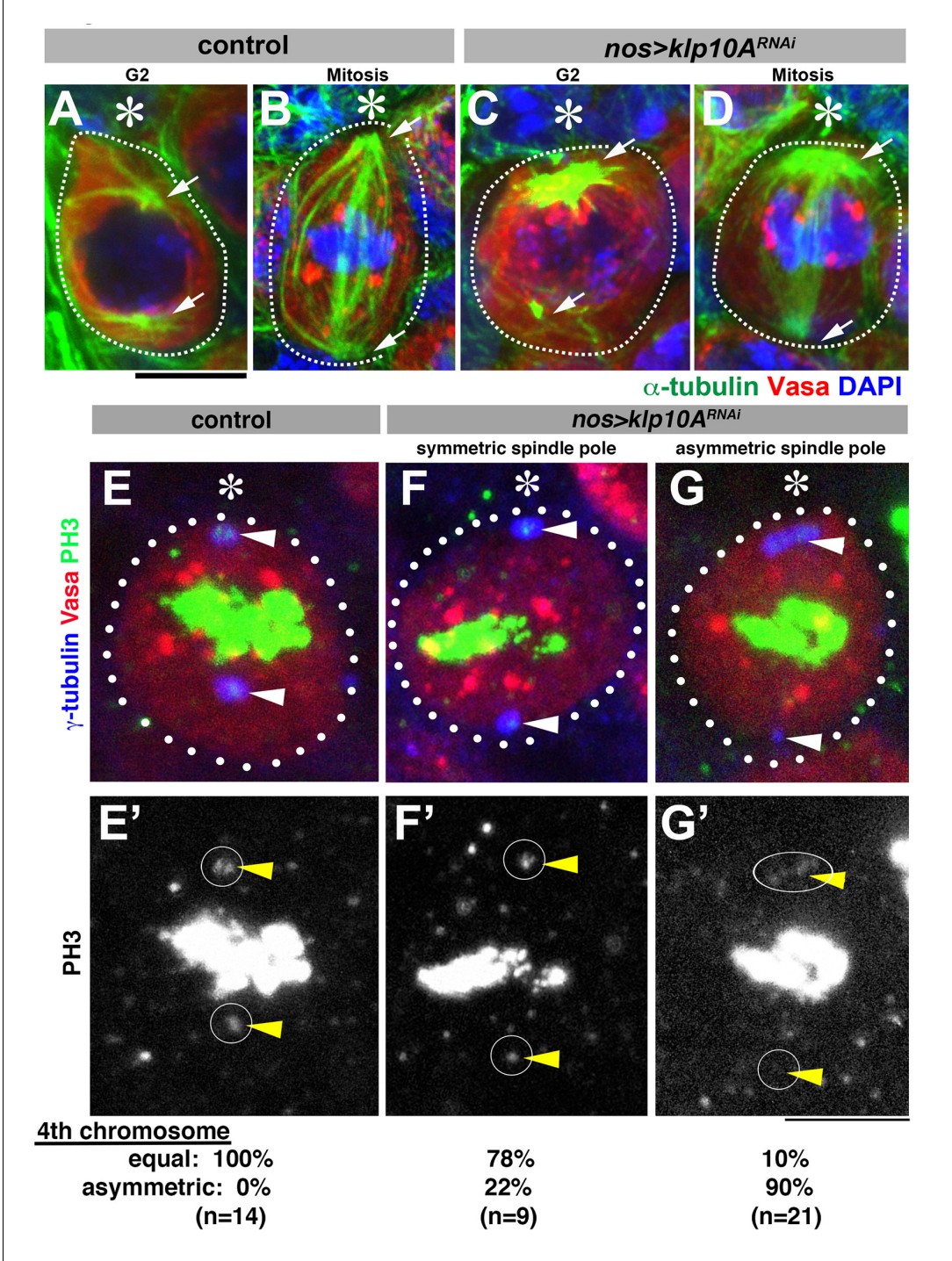

**Figure 3.** Depletion of *klp10A* results in unequal microtubule-organizing activity of the mother and daughter centrosomes. (A–D) GSCs stained for α-tubulin (green), Vasa (red) and DAPI in control G2 phase (A), control mitosis (B), *klp10A*[RNAi] G2 phase (C) and *klp10A*[RNAi] mitosis (D). GSCs are indicated by dotted lines. Centrosomes/spindle poles are indicated by arrowheads. Hub (*). Bar: 5 μm. (E–G) Mitotic GSCs stained for Vasa (red), γ-tubulin (blue), and phosphor-histone H3 (PH3, green) in control GSC (E, E'), *klp10A*[RNAi] GSC with centrosomes of equal size (F, F'), and *klp10A*[RNAi] GSC with elongated mother centrosome (G, G'). The position of spindle pole, where chromosome 4 should be observed during metaphase, is indicated by arrowheads and circles.

mitotic GSCs, *Figure 3E*). In contrast, in 90% of *klp10A^RNAi* metaphase GSCs with asymmetric centrosome size, PH3 signal was observed only near the elongated centrosomes, whereas the shorter centrosomes were not associated with any PH3 signal (*Figure 3G*, n = 21 mitotic GSCs). Although it remains possible that missegregation of chromosome 4 is due to Klp10A's function on the kinetochore, the following observation favors the possibility that asymmetric spindle force is the major cause of chromosome 4 missegregation: we observed that the frequency of chromosome 4 missegregation was considerably lower in *klp10A^RNAi* metaphase GSCs, when the two centrosomes appear to be similar in size (*Figure 3F*, 22%, n = 9 mitotic GSCs). Chromosome 4 missegregation does not likely have immediate effects on GSC viability, because chromosome 4 is essential only for development, not cell viability (*Gelbart, 1974*). These results indicate that abnormally elongated mother centrosomes in *klp10A^RNAi* GSCs cause asymmetric force generation during GSC mitoses. Consistent with the fact that we did not observe any centrosome elongation in *klp10A^RNAi* SGs, SG spindles did not show asymmetry during mitosis (not shown). Taken together, these data suggest that Klp10A is required to prevent abnormal asymmetry in MTOC activity during GSC division.

## Klp10A is required to prevent asymmetric daughter cell size

What is the consequence of mother centrosome elongation and the formation of asymmetric mitotic spindles in GSCs? By time-lapse live observation of GSCs expressing GFP-α-tubulin, we followed mitosis in control and *klp10A^RNAi* GSCs. In control GSCs, GSCs established a spindle that is symmetric in shape (the two half spindles are equal in size) and oriented toward the hub cells, as reported previously (*Figure 4A*) (*Yamashita et al., 2003*). Mitosis in control GSCs resulted in a GSC and a GB of equal size. In contrast, in *klp10A^RNAi* GSCs, abnormally elongated mother centrosomes and the normal-sized daughter centrosomes organized an asymmetric spindle, with the proximal half being much larger than the distal half (*Figure 4B*). Such asymmetry persisted to the end of mitosis, resulting in asymmetric cell size with GSCs being significantly larger than their sister GBs (*Figure 4C*).

Concomitantly with GSC division yielding two daughter cells of different cell size, we observed a striking increase in GB death. Previously, we have shown that there is a basal level of germ cell death under normal conditions (i.e. with optimal nutrient conditions in young flies) (*Yang and Yamashita, 2015*). Germ cell death is characterized by acidification of dying cells that is detectable by Lysotracker (*Yacobi-Sharon et al., 2013*; *Yang and Yamashita, 2015*). In control testes, most cell death was observed at the 16-cell stage of SGs, whereas GBs were rarely observed to die (*Figure 4D,F*) (*Yang and Yamashita, 2015*). In contrast, in *klp10A^RNAi* testis, there was a marked increase in the death of GBs (*Figure 4E,F*). This might be because GBs that are too small cannot maintain their viability, and undergo cell death. Although it is difficult to establish a causative relationship between small GB size and increased GB death, the fact that *klp10A^RNAi* increases cell death and decreases cell size specifically in GBs (but not GSCs or SGs) (*Figure 4C,F*) implies such a relationship. In support of this idea, time-lapse live observation captured an example of GB death following GSC division, where a small GB eventually lost nuclear integrity (*Figure 4—figure supplement 1*).

To explore the underlying cause(s) of GB death in *klp10A^RNAi* GSCs, we examined the segregation of major cellular components: chromosomes, Golgi and mitochondria. As described above, sister chromatids of fourth chromosomes were missegregated during mitoses of *klp10A^RNAi* GSCs (*Figure 3E–G*). However, we did not detect missegregation of major chromosomes (X, Y, second and third chromosomes). By employing FISH method using chromosome specific probes, we found no evidence for chromosome missegregation for major chromosomes (*Figure 4—figure supplement 2*). Therefore, it is unlikely that the chromosome missegregation is the cause of GB death. Next we examined the segregation patterns of cellular organelles. Mitochondria segregated equally during GSC division in control/wild type, whereas more mitochondria were inherited by GB in *klp10A^RNAi* testes (*Figure 5A–C*). Because GBs inherited more mitochondria than GSCs, it is unlikely that missegregation of mitochondria explains the higher frequency of GB death. It is of note that the GBs, which have less MTOC activity, inherit more mitochondria. The underlying mechanism of this phenomenon is currently unclear. Golgi was observed to be inherited predominantly by GBs during control/wild type GSC divisions, whereas GSCs inherited more Golgi upon knockdown of *klp10A* (*Figure 5D,E*). If the amount of Golgi is essential for GB viability, this might explain why GBs die more frequently. In summary, although it remains unclear whether the perturbed segregation patterns of any single organelle can explain the frequent GB death, these data suggest that asymmetric spindle in *klp10A^RNAi* GSCs results in altered patterns of organelle segregation, which might

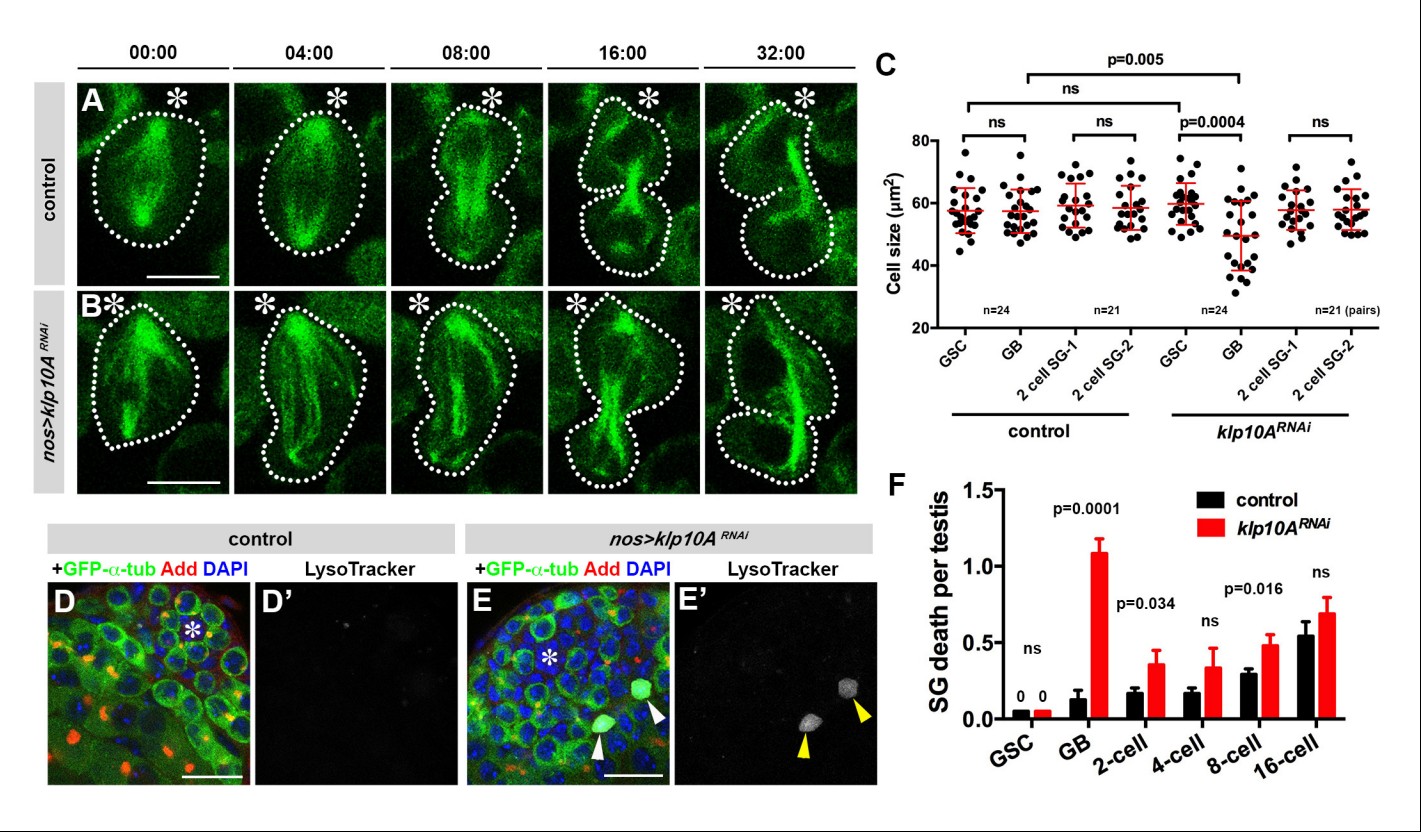

**Figure 4.** Depletion of *klp10A* generates GSC daughters with unequal cell sizes. (**A, B**) Frames from time-lapse live observation of control (**A**) and *klp10A^RNAi* (**B**) GSCs expressing GFP-α-tubulin. GSCs are outlined by dotted lines. Time is indicated in minutes. Movies were started during late metaphase, ~4 min prior to spindle elongation. Hub (*).Bar: 5 μm. (**C**) Quantification of cell size in GSCs and GBs or two SGs at the completion of GB mitosis to become 2-cell SGs in control vs. *klp10A^RNAi* testes. N = 24 mitotic GSCs and N = 21 mitotic GBs were scored in both control and *klp10A^RNAi* testes. (**D, E**) testis apical tip stained for GFP-α-tubulin (green), Adducin-like (red), DAPI (blue) and Lysotracker (white). Hub (*). Dying GB is indicated by arrowheads in E). Bar: 20 μm. F) Quantification of germ cell death in control vs. *klp10A^RNAi* testes. N = 96 testes were scored in both control and *klp10A^RNAi* for quantification. p value was obtained with Student's t-test (two-tailed). Data are shown as mean ± s.d.

The following figure supplements are available for figure 4:

**Figure supplement 1.** An example of GB death following *klp10A^RNAi* GSC division with unequal daughter cell size.

**Figure supplement 2.** Segregation of major chromosomes is not affected in *klp10A^RNAi* testes.

collectively compromise the viability of GBs. Alternatively, a cell size checkpoint may be in operation, possibly eliminating cells that are too small (*Kellogg, 2003*). It is also possible that a cell competition mechanism might lead to elimination of unfit cells (*Johnston, 2009*).

## Discussion

Unique behaviors of centrosomes in stem cell populations have been reported in multiple stem cell systems (*Conduit and Raff, 2010*; *Habib et al., 2013*; *Januschke et al., 2011*, *2013*; *Rebollo et al., 2007*; *Rusan and Peifer, 2007*; *Salzmann et al., 2014*; *Wang et al., 2009*; *Yamashita et al., 2003*, *2007*). Asymmetric behavior of the mother and daughter centrosomes in these stem cell populations further led to the speculation that the mother and daughter centrosomes might be different not only from each other but also different from the centrosomes in non-stem cells, potentially contributing to asymmetric cell fate determination (*Pelletier and Yamashita, 2012*; *Tajbakhsh and Gonzalez,*

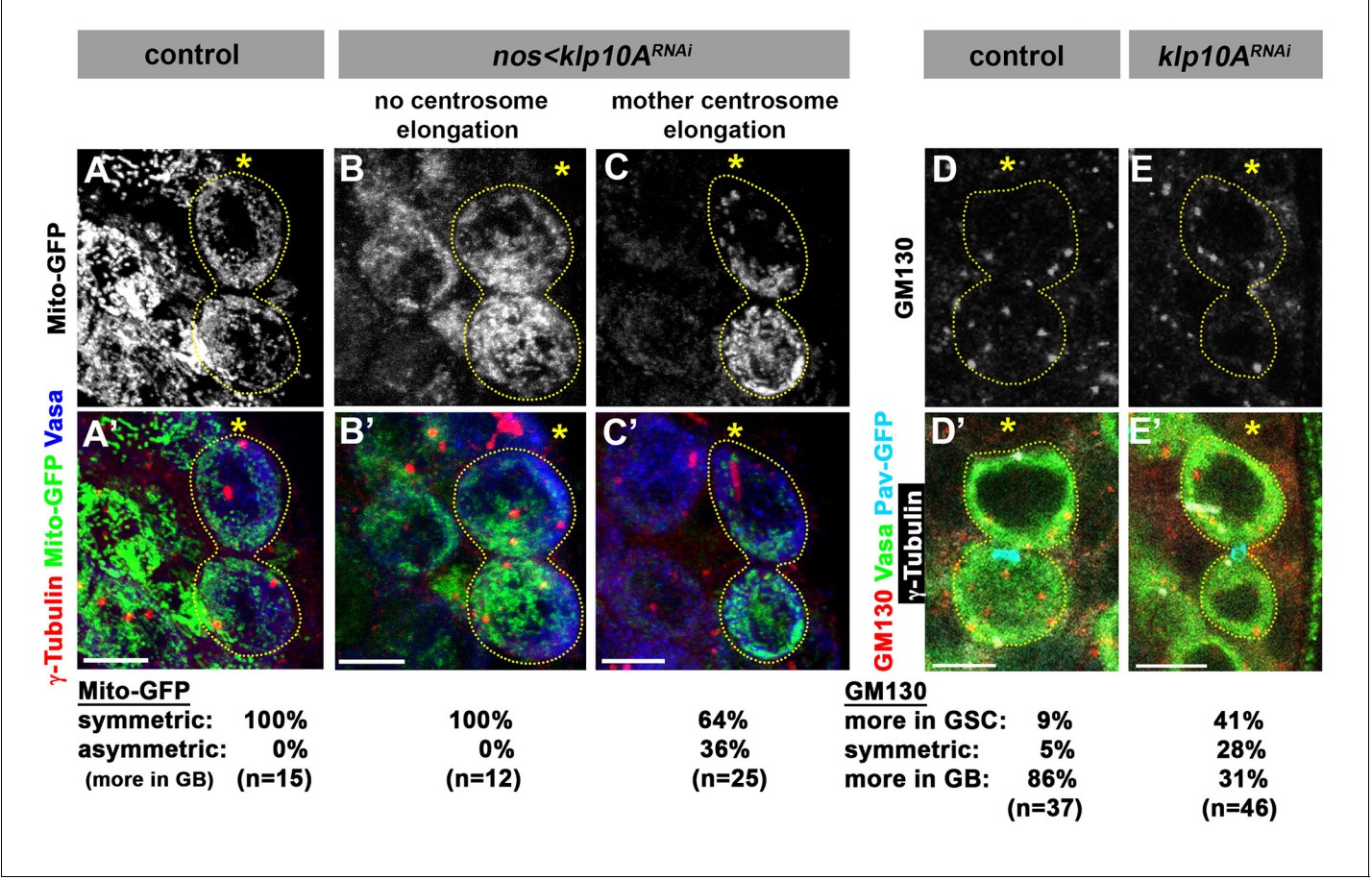

**Figure 5.** Segregation of mitochondria and Golgi is affected in *klp10A^RNAi* testes. (A–C) Mito-GFP distribution in control (A) and *klp10A^RNAi* (B, C) GSC-GB pairs. Red: γ-tubulin. Green: mito-GFP. Blue: Vasa. (D, E) GM130 (Golgi) distribution in control (D) and *klp10A^RNAi* (E) GSC-GB pairs. Asterisks indicate the hub, and dotted lines indicate GSC-GB pairs. Bars: 5 μm.

2009). However, it remained unknown whether stem cell centrosomes are indeed unique, and whether the stem cell mother centrosome is distinct from the daughter centrosome.

In this study, we identified Klp10A as the first protein that is enriched in interphase centrosomes specifically in stem cells. Its localization and mutant phenotype provide important insights into centrosome biology of stem cells. Klp10A's enrichment on the stem cell centrosomes but not on the centrosomes of differentiating SGs suggests the presence of cellular mechanisms unique to stem cell centrosomes. Moreover, *klp10A* knockdown/mutation led to elongation of the mother centrosome specifically within stem cells; neither the daughter centrosome in GSCs nor any centrosomes in SGs were elongated. This phenotype demonstrates that the stem cell mother centrosome has unique characteristics and modes of regulation and indicates that the GSC mother centrosome has an inherent tendency to elongate, which must be counteracted by Klp10A. The mother centrosome in GSCs may have distinct microtubule dynamics, partly because of the necessity for the mother centrosome to be anchored to the hub-GSC interface throughout the cell cycle (*Yamashita et al., 2007*). It is worth noting that the centrosome size itself might not be the cause of asymmetric force generation. It was recently reported that loss of function of *asterless (asl)*, the gene that encodes a core centriolar component, results in mother centrosome-specific elongation in *Drosophila* male GSCs (*Galletta et al., 2016*). However, *asl* mutant did not lead to asymmetric spindle formation or daughter cell size, suggesting a unique function for Klp10A.

Asymmetric spindles, where one half of the spindle is considerably larger than the other, are often observed during normal developmental processes. This results in asymmetry in cell sizes as

well. For example, *Drosophila* neuroblasts divide asymmetrically to produce another neuroblast and a differentiating cell (either ganglion mother cell in type I neuroblast lineage, or intermediate neural progenitor cell in type II neuroblast lineage) (*Doe and Bowerman, 2001*; *Prehoda, 2009*). In neuroblast divisions, the self-renewing neuroblast is considerably larger than its differentiating daughter cell. Similarly, the first mitosis of *C. elegans* zygote is also characterized by asymmetric daughter cell sizes (*Rose and Gonczy, 2014*). Asymmetric spindle morphology contributes to cell size asymmetry together with a myosin-dependent pathway that controls cleavage furrow positioning (*Cabernard et al., 2010*; *Knoblich, 2010*; *Pacquelet et al., 2015*). These examples highlight that the asymmetry in daughter cell sizes can be a programmed process and does not inherently lead to cell death, as observed in *klp10A*$^{RNAi}$ GSC divisions. More interestingly, in *C. elegans* Q lineage neuroblasts, which divide asymmetrically by spindle pulling and asymmetric Myosin localization, the smaller daughter undergoes apoptosis reminiscent of GBs in *klp10A*$^{RNAi}$ testes (*Ou et al., 2010*). This implies that death of daughter cells that are born too small might be utilized in the context of normal development. Our observations described in this study suggest that multiple aspects of ACDs (e.g. asymmetric segregation of fate determinants, spindle orientation, daughter cell size asymmetry) are carefully calibrated processes to achieve successful ACD unique to individual systems.

In summary, our study illuminates a critical mechanism for ACD, in which the MT-depolymerizing kinesin, Klp10A, counterbalances the mechanism that generates asymmetric centrosome behavior of mother and daughter centrosomes. Whereas asymmetry-generating mechanisms are critical for proper spindle orientation and thus ACD of GSCs, the lack of a counteracting force by Klp10A results in excessive asymmetries, leading to frequent death of GBs. We propose that ACD is achieved by fine-tuned symmetries and asymmetries, failure of which can lead to unsuccessful cell divisions.

## Materials and methods

### Fly husbandry and strains

All fly stocks were raised in standard Bloomington medium. *ubi-asl-YFP* (*Varmark et al., 2007*), *cnb-YFP* (*Januschke et al., 2011*) (a gift of Cayetano Gonzalez, IRB Barcelona), *nos-gal4* (*Van Doren et al., 1998*), *UAS-klp10A-GFP* (*Inaba et al., 2015*), *UAS-klp10A*$^{TRiP.HMS00920}$ (obtained from the Bloomington Stock Center), *UAS-α-tubGFP* (*Grieder et al., 2000*) (a gift of Allan Spradling, Carnegie Institution of Washington), and *UAS-mito-GFP* (obtained from the Bloomington Stock Center) (*Cox and Spradling, 2003*).

### Immunofluorescence staining, fluorescence in situ hybridization (FISH) and microscopy

Immunofluorescence staining was conducted as described previously (*Cheng et al., 2008*). To better preserve microtubules during fixation (*Figure 3*), testes were dissected in 60 mM PIPES, 25 mM HEPES, 10 mM EGTA, 4 mM MgSO$_4$, pH 6.8 (PEM buffer) and fixed in 4% formaldehyde in PEM for 12 min. Samples were washed in phosphate-buffered saline (PBS) containing 0.3% Triton X-100 (PBST) overnight at 4°C, followed by incubation with primary antibodies at 4°C overnight. Samples were washed for 60 min (three 20 min washes) in PBST at 25°C, incubated with Alexa Fluor-conjugated secondary antibodies in PBST containing 3% bovine serum albumin (BSA) at 25°C for 2 hr, washed as above, and mounted in VECTASHIELD with DAPI (Vector Labs, Burlingame, CA).

The primary antibodies used were: mouse anti–fasciclin III (1:20; developed by C. Goodman [*Patel et al., 1987*]), rat anti-Vasa (1:20; developed by A. Spradling and D. Williams), mouse anti–Adducin-like (1:20; developed by H. D. Lipshitz [*Ding et al., 1993*]), and mouse anti–α-tubulin (4.3; 1:50; developed by C. Walsh [*Walsh, 1984*]) (obtained from the Developmental Studies Hybridoma Bank), rabbit anti-Vasa (d-26; 1:200; Santa Cruz Biotechnology, Santa Cruz, CA), mouse anti–γ-tubulin (GTU-88; 1:100; Sigma-Aldrich, St. Louis, MO), rabbit anti–Thr 3-phosphorylated histone H3 (1:200; EMD Millipore, Germany), and rabbit anti-Klp10A (*Rogers et al., 2004*) (1:2000, a gift from David Sharp). Guinea pig anti-Klp10A (1: 1000) was generated using the same antigen used to generate rabbit anti-Klp10A antibody (Covance, Princeton, NJ). Specificity of the serum was tested by

western blotting and by immunohistochemistry on Klp10A-depleted testes. Alexa Fluor-conjugated secondary antibodies (Thermo Fisher Scientific, Waltham, MA) were used at a dilution of 1:200.

For FISH analysis, testes from 0–3 days-old adult flies were dissected in 1X PBS, fixed for 30 min with 4% formaldehyde in PBS + 1 mM EDTA, permeabilized for 50 min in PBST (0.1% Triton X-100 in PBS)+1 mM EDTA and incubated in 3% BSA, PBST+EDTA overnight at 4°C with the primary antibodies: rabbit anti-Vasa (1:200; Santa Cruz Biotechnology), mouse anti-Adducin-like and mouse anti-Fasciclin III. Samples were washed with PBST + 1 mM EDTA for 50 min and incubated overnight at 4°C with Alexa Fluor 594- and 488-conjugated secondary antibodies (1:200). After washing with PBST + EDTA for 30 min, samples were fixed for 10 min with 4% formaldehyde in PBS + 1 mM EDTA followed by wash in PBST + 1 mM EDTA for 30 min and rinsed with PBST. Samples were treated with RNase A (2 mg/ml in water) for 10 min at 37°C and washed with PBST + 1 mM EDTA for 10 min. In preparation for FISH, the buffer was gradually exchanged to 50% formamide, 2xSSC + 1 mM EDTA, with steps of 0%, 20%, 40% and 50% formamide concentration (each step 10 min incubation). Hybridization with Cy-3- and Cy-5-labeled ssDNA probes (1 µM end concentration) has been performed in a buffer of 50% formamide, 10% dextran sulfate and 2xSSC + 6.5 mM EDTA overnight at 37°C immediately after a 2 min denaturation step at 90°C. After hybridization, 1 hr wash in 2xSSC +EDTA has been performed and testes were mounted in VECTASHIELD+DAPI (H-1200, Vector Laboratories). The following chromosome specific probes were used: Cy3-(AATAC)$_6$ or Cy5-(GTATT)$_6$ to detect Y chromosome, Cy3-CCACATTTTGCAAATTTTGATGACCCCC CTCCTTACAAAAAATGCG to detect X chromosome, Cy3-(AACAC)$_6$ to detect second chromosome, and Cy3- ACCGAGTACGG-GACC GAGTACGGGACCAGTAC to detect third chromosome (based on (*Yadlapalli and Yamashita, 2013*). Note that the probes for second and third chromosomes also hybridize to Y chromosome.

Images were taken using a Leica TCS SP5 or a Leica TCS SP8 confocal microscope with a 63× oil-immersion objective (NA = 1.4) and processed using Adobe Photoshop software.

## Time-lapse live imaging

Testes from newly eclosed flies were dissected into Schneider's *Drosophila* medium containing 10% fetal bovine serum. The testis tips were placed inside a sterile glass-bottom chamber and were mounted on a three-axis computer-controlled piezoelectric stage. An inverted Leica TCS SP8 confocal microscope with a 63× oil immersion objective (NA = 1.4) was used for imaging and images were processed using Adobe Photoshop software. Cell size analysis following GSC and GB mitosis was determined using area ratio of two daughter cells by manually drawing regions of interest in ImageJ.

## Quantification of cell death

For detection of germ cell death, testes were stained with Lysotracker Red DND-99 (Thermo Fisher Scientific) in PBS for 30 min prior to formaldehyde fixation. Anti-Adducin staining (described above) was used to distinguish various SG stage. Dying SGs were scored according to stage by counting the number of Lysotracker+ germ cell nuclei within a cyst.

## Transmission electron microscopy

*Drosophila* testes were dissected in 1× PBS and fixed in 2% glutaraldehyde/2% paraformaldehyde (EM grade) in 0.1 M cacodylate (pH 7.4) for 5 min at room temperature. This step was followed by an additional 25 min fixation on ice. The tissue was rinsed three times for 10 min each in cacodylate buffer, and then post-fixed for 30 min in 2% osmium tetroxide in the same buffer on ice. Next, the samples were rinsed in double-distilled water, and then *en bloc* stained for one hour in aqueous 7% uranyl acetate. The samples were then dehydrated in increasing concentrations of ethanol, treated with propylene oxide, and embedded in Epon epoxy resin. Semi-thin sections were stained with toluidine blue for tissue identification. Selected regions of interest were serial-sectioned (70 nm thickness) and mounted on Formvar/carbon-coated slotted grids. The grids were post-stained with uranyl acetate and lead citrate, and samples were examined using a Philips CM100 electron microscope at 60 KV. Images were recorded digitally using a Hamamatsu ORCA-HR digital camera system, which was operated using AMTsoftware (Advanced Microscopy Techniques Corp., Danvers, MA).

## Acknowledgements

We thank Drs. Cayetano Gonzalez, David Glover, David Sharp, the Bloomington Stock Center, and the Developmental Studies Hybridoma Bank for reagents; the Microscopy Imaging Laboratory at the University of Michigan for electron microscopy; Drs. Nasser Rusan, Clemens Cabernard, Tomer Avidor-Reiss, Shukry Habib and the Yamashita laboratory members for critical reading of the manuscript; and Dr. Nasser Rusan for sharing unpublished results. This research is supported by Howard Hughes Medical Institute, and partly by National Institute of General Medical Sciences (R01GM118308 to YY). YY is supported by the John D and Catherine T MacArthur Foundation.

## Additional information

### Competing interests

YMY: Reviewing editor, *eLife*. The other authors declare that no competing interests exist.

### Funding

| Funder | Grant reference number | Author |
| --- | --- | --- |
| Howard Hughes Medical Institute | | Yukiko M Yamashita |
| National Institute of General Medical Sciences | R01GM118308 | Yukiko M Yamashita |

The funders had no role in study design, data collection and interpretation, or the decision to submit the work for publication.

### Author contributions

CC, Designed and conducted experiments (Figure 4, 5), Interpreted data, Edited the manuscript, Conception and design, Acquisition of data, Analysis and interpretation of data, Drafting or revising the article; MI, Designed and conducted experiments (Figure 1, 2), Interpreted data, Edited the manuscript, Conception and design, Acquisition of data, Analysis and interpretation of data, Drafting or revising the article; ZGV, Designed and conducted experiments (Figure 3, 5), Interpreted data, Edited the manuscript, Conception and design, Acquisition of data, Analysis and interpretation of data, Drafting or revising the article; YMY, Designed and conducted experiments (Figure 3), Interpreted data, Wrote the manuscript, Conception and design, Acquisition of data, Analysis and interpretation of data, Drafting or revising the article

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
