## [Decision Letter]

Thank you for submitting your article "Klp10A, a stem cell-specific centrosomal kinesin, balances asymmetries in *Drosophila* male germline stem cell division" for consideration by *eLife*. Your article has been reviewed by two peer reviewers, and the evaluation has been overseen by K VijayRaghavan as the Senior Editor and Reviewing Editor. The following individuals involved in review of your submission have agreed to reveal their identity: Clemens Cabernard (Reviewer #1) and Tomer Avidor-Reiss (Reviewer #2).

The reviewers have discussed the reviews with one another and the Reviewing Editor has drafted this decision to help you prepare a revised submission.

Summary:

Asymmetric cell division can be manifested in the biased segregation of cell fate determinants and organelles or cell size differences between daughter cells. Recent observations have shown that centrosomes are intrinsically asymmetric and can also segregate asymmetrically. For instance, in male germline stem cells, centrosomes containing the old mother centriole remain in the stem cell whereas the daughter-centriole containing centrosome segregates into the differentiating gonialblasts. It was proposed that centrosomes contain stem cell specific factors, contributing to cell fate decisions. However, no such factors have currently been identified.

In the manuscript by Cuie Chen et al., the authors report the identification of Klp10A, a microtubule-depolymerizing kinesin of the kinesin-13 family, as a cell-specific centrosomal protein in *Drosophila* male germline stem cells (GSCs). The authors claim that Klp10A is predominantly localized to mother centrosomes in interphase GSCs. Loss of function experiments further revealed that mother centrosomes elongate, specifically in GSCs. This elongation induces asymmetric spindles in mitotic GSCs, causing physical asymmetric divisions. The authors further observe an increase in cell death and propose that physical asymmetric GSC divisions produce smaller than normal differentiating cells which die more frequently.

Essential revisions:

This is an interesting paper, that could potentially provide novel insight into centrosome asymmetry and basic stem cell biology. However, we recommend that the authors carefully address the following critiques:

1) We are not entirely convinced about the authors claim that Klp10A is a GSC specific centrosomal protein. As Figure 1 shows, other non-GSC cells also show Klp10A localization on centrosomes.

2) The authors are using RNAi and a hypomorphic allele to remove Klp10A and claim that the RNAi line almost completely abolishes Klp10A protein expression. However, it is very difficult to judge from Figure 2—figure supplement 1 whether Klp10A is efficiently removed due to a considerable background signal. An ideal experiment would be to show this in RNAi clones, allowing the comparison between wild type and RNAi expressing GSCs. Alternatively, high magnification images showing Klp10A expressing GSCs stained for Klp10A and the centriolar marker Asl should be provided.

3) The authors claim that Klp10A specifically prevents centrosome elongation in GSCs. However, in spermatocytes, centrosomes also elongate, albeit not restricted to mother centrosomes. It is not quite clear why the authors maintain that Klp10A is GSC specific (see also (4) below).

4) To show that Klp10A specifically affects mother centrosomes, Chen et al. express PACT-GFP, observing that elongated centrosomes predominantly carry this label. However, an alternative explanation could be that this centrosomal marker does not discriminate between mother and daughter centrosome but simply incorporates into the biggest centrosome. In wild type GSCs centrosome size might correlate with centrosome age, however, that does not have to be the case in Klp10A mutants. Thus, although likely, it has not been formally proven that Klp10A specifically compromises mother centrosomes. Due to the lack of mother centrosome specific markers, it might be best change the wording accordingly.

5) It is unclear if the essential activity of Klp10A in the centrosome mediates the other activities during mitosis, or if Klp10A has multiple independent sites of action that mediate the various activities. In this regard, it would be useful to identify a site on Klp10A that controls its localization/enrichment to the GSC centrosome (but not to the spindle) and test if the centrosome localization is critical to all the reported activities of Klp10A. This type of experiment would allow substantiation of the claim that "Abnormal elongation of the mother centrosome leads to asymmetric spindle formation, which in turn results in asymmetric cell size." Without this type of an experiment, this claim would need to be deleted.

6) Given the striking nature of the loss of function phenotype, it might be interesting to test whether over-expressing Klp10A is sufficient to decrease centrosome size. The authors may want to see if this suggested experiment can be done within a reasonable (<2months) time.

7) We are convinced by the observation that mitotic GSCs devoid of Klp10A show asymmetric MTOC activity, resulting in asymmetric spindles. However, the live cell imaging shows that once cytokinesis is completed, both sibling cells seem to be similar in cell size, comparable to wild type. Furthermore, the authors claim that gonialblasts are smaller in size after knocking-down Klp10A. The authors only provide ratio's but not absolute cell size values that could be compared to wild type. Thus, it is not clear whether GBs indeed are smaller or if GSCs are bigger compared to wild type. To support their claim, the authors should provide cell size or cell diameter measurements.

8) Also, whether cell size indeed correlates with cell viability is not shown. To that end, the authors would have to track the fate of GBs, resulting from asymmetric GSC divisions. We are not sure this is an experiment that is technically feasible in this system. Thus, it remains unclear whether the increase in cell death is indeed a consequence of reduced gonialblast size.

9) In the Discussion, the authors claim that cell size asymmetry is dictated by spindle morphology. This is not entirely true since studies in *Drosophila* and *C. elegans* neuroblasts have shown that Myosin localization is also an important determinant of sibling cell size. Thus, this statement should be corrected.

---

## [Author Response]

***[…]***

*Essential revisions:*

*This is an interesting paper, that could potentially provide novel insight into centrosome asymmetry and basic stem cell biology. However, we recommend that the authors carefully address the following critiques:*

*1) We are not entirely convinced about the authors claim that Klp10A is a GSC specific centrosomal protein. As Figure 1 shows, other non-GSC cells also show Klp10A localization on centrosomes.*

We edited the figure to clarify this point by adding a separate channel for Asl-YFP. Indeed, Klp10A’s localization to interphase centrosome is specific to GSCs. In non-stem cells, Klp10A localizes to the microtubule bundle in telophase and mitotic spindle poles, but not interphase centrosomes. We also clarified this in the text.

*2) The authors are using RNAi and a hypomorphic allele to remove Klp10A and claim that the RNAi line almost completely abolishes Klp10A protein expression. However, it is very difficult to judge from Figure 2—figure supplement 1 whether Klp10A is efficiently removed due to a considerable background signal. An ideal experiment would be to show this in RNAi clones, allowing the comparison between wild type and RNAi expressing GSCs. Alternatively, high magnification images showing Klp10A expressing GSCs stained for Klp10A and the centriolar marker Asl should be provided.*

We apologize for the confusion. We have published the validation of klp10A-RNAi already (Inaba et al., 2015 Nature). But in the Inaba paper, we did not have a space to show the actual data to demonstrate efficient knockdown. Thus, we added such data (Klp10A-RNAi germline clone stained with anti-Klp10A), which clearly demonstrate efficient knockdown (Figure 2—figure supplement 1), and also cited the Inaba (2015).

*3) The authors claim that Klp10A specifically prevents centrosome elongation in GSCs. However, in spermatocytes, centrosomes also elongate, albeit not restricted to mother centrosomes. It is not quite clear why the authors maintain that Klp10A is GSC specific (see also (4) below).*

We agree with the reviewers. Our intention was 1) centrosome elongation is considerably more dramatic in GSCs (compared to spermatocytes), and 2) Klp10A mutant would have ‘asymmetric centrosome’ problem only in GSCs (but not in spermatocytes, because both centrosomes elongates in spermatocytes). We clarified this point in the revised text not to indicate that Klp10A’s expression and function is limited to GSCs.

*4) To show that Klp10A specifically affects mother centrosomes, Chen et al. express PACT-GFP, observing that elongated centrosomes predominantly carry this label. However, an alternative explanation could be that this centrosomal marker does not discriminate between mother and daughter centrosome but simply incorporates into the biggest centrosome. In wild type GSCs centrosome size might correlate with centrosome age, however, that does not have to be the case in Klp10A mutants. Thus, although likely, it has not been formally proven that Klp10A specifically compromises mother centrosomes. Due to the lack of mother centrosome specific markers, it might be best change the wording accordingly.*

Thanks for the suggestion. We modified the text accordingly.

(We added a sentence: “Although we cannot exclude the possibility that PACT is preferentially incorporated into the larger centrosome (irrespective of age) in *klp10A^RNAi^*GSCs, the above data indicate that it is likely the mother centrosome that elongates upon loss of *klp10A* function.”).

*5) It is unclear if the essential activity of Klp10A in the centrosome mediates the other activities during mitosis, or if Klp10A has multiple independent sites of action that mediate the various activities. In this regard, it would be useful to identify a site on Klp10A that controls its localization/enrichment to the GSC centrosome (but not to the spindle) and test if the centrosome localization is critical to all the reported activities of Klp10A. This type of experiment would allow substantiation of the claim that "Abnormal elongation of the mother centrosome leads to asymmetric spindle formation, which in turn results in asymmetric cell size." Without this type of an experiment, this claim would need to be deleted.*

We fully agree with this comment. We modified the text accordingly. (Indeed, it is of our future interest to study localization specific function of Klp10A. However, such experiments cannot be done in a timely manner, especially because there is a possibility that a Klp10A mutant protein that is defective in localizing to one location (e.g. centrosome) might fail to localize another location (e.g. kinetochore)). However, please note that asymmetric spindle formation correlated very well with mother centrosome elongation (on the contrary, even when klp10A knockdown is complete, if the mother centrosome has not elongated, such GSCs did not lead to asymmetric spindle formation). Considering these points, we modified the text to clarify that 1) we do not have sufficient data to attribute all klp10A mutant phenotype to Klp10A’s function on the centrosome, and 2) yet mother centrosome elongation correlate well with the observed phenotype.

Now the relevant part of the Abstract reads: Depletion of *klp10A* results in abnormal elongation of the mother centrosomes in GSCs, suggesting the existence of a stem cell-specific centrosome regulation program. Concomitant with mother centrosome elongation, GSCs form asymmetric spindle, wherein the elongated mother centrosome organizes considerably larger half spindle than the other half. This leads to asymmetric cell size, yielding a much smaller differentiating daughter that dies more frequently.

We also modified the text throughout to reflect these notions.

*6) Given the striking nature of the loss of function phenotype, it might be interesting to test whether over-expressing Klp10A is sufficient to decrease centrosome size. The authors may want to see if this suggested experiment can be done within a reasonable (<2months) time.*

We have done this, and observed no shortening of the centrosome upon overexpression of the klp10A. 0.69 ± 0.26(sd) µm in control GSCs vs. 0.71 ± 0.22 µm in klp10A-RNAi GSCs (N=67 GSCs for each genotype, p=0.70). (the same UAS-Klp10A expression construct perfectly rescues centrosome elongation phenotype of Klp10A-RNAi, demonstrating that this construct certainly express functional Klp10A). We incorporated this information in the revised text.

*7) We are convinced by the observation that mitotic GSCs devoid of Klp10A show asymmetric MTOC activity, resulting in asymmetric spindles. However, the live cell imaging shows that once cytokinesis is completed, both sibling cells seem to be similar in cell size, comparable to wild type. Furthermore, the authors claim that gonialblasts are smaller in size after knocking-down Klp10A. The authors only provide ratio's but not absolute cell size values that could be compared to wild type. Thus, it is not clear whether GBs indeed are smaller or if GSCs are bigger compared to wild type. To support their claim, the authors should provide cell size or cell diameter measurements.*

We have re-made the graph to show actual GSC/GB sizes, instead of the ratio. Please note that panel C is GSC-GB size when they are clearly in the next G1 phase. Although GB seems to grow back in G1 in panel B (and GSC-GB size difference seems to be more striking in telophase), the GSC-GB size were measured in next G1 (after clear completion of mitosis), and panel C reflects the measurement in the next G1 phase.

Indeed, we think that some GB can grow back to a (semi)normal size, and not all of them die. We have conducted live observation to follow GSC-GB fate following their division with unequal cell size. GSC cell cycle is 12-16 hours based on several studies (including ours, Yadlapalli 2011 JCS, Inaba 2015 *eLife*). Thus, following cell division and subsequent GB fate in the time frame of live observation (up to 20 hours) is somewhat challenging. However, out of 10 such movies, we captured one example, where GB lost nuclear integrity and apparently started dying. Now we incorporate this data as supplementary figure (Figure 4—figure supplement 1).

*8) Also, whether cell size indeed correlates with cell viability is not shown. To that end, the authors would have to track the fate of GBs, resulting from asymmetric GSC divisions. We are not sure this is an experiment that is technically feasible in this system. Thus, it remains unclear whether the increase in cell death is indeed a consequence of reduced gonialblast size.*

As described above, now we add a result of live observations. However, challenging nature of live observation to track GSC-GB division AND subsequent GB fate (death) makes it difficult to ‘correlate’ GB size and their death. Therefore, we modified the text not to mislead the readers, and more modestly stated our conclusion.

*9) In the Discussion, the authors claim that cell size asymmetry is dictated by spindle morphology. This is not entirely true since studies in Drosophila and C. elegans neuroblasts have shown that Myosin localization is also an important determinant of sibling cell size. Thus, this statement should be corrected.*

Agreed. In the original version, we were simply making a comparison between neuroblasts and klp10A mutant GSCs regarding spindle-mediated cell size asymmetry (which were not a complete description of the knowledge in the field). In the revised version, we clearly stated that spindle asymmetry is not the only mechanism to account for asymmetric daughter cell sizes and that myosin localization contributes to cell size asymmetry.